

# Tropical Cyclone Asymmetric Eyewall Evolution and Intensification in a Two-Layer Model

Ting-Yu Cha[1,2] and Michael M. Bell[2]

[1]National Center for Atmospheric Research, Boulder, Colorado, United States
[2]Department of Atmospheric Science, Colorado State University, Fort Collins, Colorado, United States

**Correspondence:** Ting-Yu Cha (tycha@ucar.edu)

**Abstract.** Radar and satellite imagery of numerous intensifying tropical cyclones (TCs) depict an appearance of polygonal eyewall structure where deep convection is often located near the polygonal vertices. A recent observational study of Hurricane Michael's (2018) polygonal eyewall evolution suggests that the vorticity asymmetries are coupled with the reflectivity asymmetries during rapid intensification. Conceptual theory of a polygonal eyewall structure has been linked to vortex Rossby waves

(VRWs) and the breakdown of an enhanced potential vorticity (PV) ring, but how the asymmetries affect TC intensification remains unclear. Non-divergent barotropic models have been previously employed to study polygonal eyewall dynamics, but this approach has limitations due to the importance of diabatic heating to PV generation and the intensification process. Results from prior studies motivate us to explore the nature of the relationship between asymmetric vorticity and vertical velocity in the free atmosphere and the boundary layer and their compound impacts on the TC intensification process. Here we use a simple

two-layer model framework with a shallow water model on top of a slab boundary layer model (SBL) to simulate a frictional boundary layer underneath the free atmosphere. Results from simulating a wavenumber-two elliptical asymmetry suggest the VRW in the free atmosphere can organize the updrafts in the SBL, which is consistent with radar observations of enhanced reflectivity at the polygonal vertices. Free atmospheric divergence in the shallow water layer does not explain the coupling between vorticity and reflectivity. The coupling can be explained to first order by the one-way boundary layer response to the

pressure gradient associated with the free atmospheric vorticity asymmetries, consistent with prior studies. Further simulations that allow two-way interaction between the layers show that the organization of the updrafts out of the SBL plays a critical role in the growth of a PV ring and intensification of the mean vortex. In this framework, diabatic heating in the shallow-water layer parameterized by a mass sink driven by the free-atmosphere/SBL interaction leads to rapid intensification of the vortex, thinning of the PV ring, and eventual barotropic instability and PV mixing. The simplified modeling framework with two-way

interactions captures many of the essential dynamics of rapid intensification in the presence of evolving asymmetries similar to those seen in the observations from Hurricane Michael (2018), which provides new insight into the complex interactions between dynamics and convection during hurricane intensification.





## 1 Introduction

Radar and satellite imagery have captured polygonal eyewall structures in numerous intensifying tropical cyclones (TCs), such

as Typhoon Herb (1996) (Kuo et al., 1999), Hurricane Isabel (2003) (Kossin and Schubert, 2004), Dolly (2008) (Hendricks et al., 2012), Karal (2010) (Guimond et al., 2016), and Michael (2018) (Cha et al., 2020). Since the precipitation structure can be organized by the underlying dynamics, numerous theoretical studies have conceptually described the dynamics behind the observed polygonal shape in reflectivity or satellite brightness temperatures as a result of barotropic or combined barotropic-baroclinic instability of a potential vorticity (PV) ring through counter-propagating vortex Rossby waves (VRWs).

The instability can further lead to the breakdown of the PV ring and eye-eyewall mixing (Kuo et al., 1999; Schubert et al., 1999; Hendricks et al., 2010). In Cha et al. (2020), it was demonstrated through analysis of observed reflectivity and retrieved tangential wind asymmetries of Hurricane Michael during rapid intensification that barotropic linear VRW theory provided a good quantitative explanation of the azimuthal propagation speed of the polygonal eyewall structure, suggesting a close coupling of the vorticity asymmetries and deep convection. Observations are able to resolve some of the convective, dynamic,

and thermodynamic characteristics, but are difficult to use alone to fully understand the key physical processes due to limited high-temporal and spatial observations of complex atmospheric structure. The mechanisms that force deep convection at the polygonal vertices and how those convective asymmetries play a role in TC intensification were unable to be addressed with the observational dataset in Cha et al. (2020).

As an alternative to the complexity of real observations, many studies employ full-physics modeling to provide four-

dimensional dynamical and thermodynamical fields to study the relationships between dynamics and convective processes. However, the results of such studies can depend on the design, initial conditions, and physics of the model, and the dynamics in full-physics models can be hard to interpret due to complex interactions among many different processes. A simpler approach to understanding the dynamics involves models with less complexity. A balanced vortex framework has been frequently employed to understand TC intensification, which assumes that an axisymmetric vortex continuously evolves in a state of gradient

wind and hydrostatic balance (Eliassen, 1951; Schubert and Hack, 1982; Willoughby, 1979; Smith et al., 2018). While an axisymmetric balanced vortex framework can offer many insights into TC intensification (Ooyama, 1969), it is unable to fully represent asymmetric mechanisms and boundary layer processes due to the inherent assumptions. In this study, we investigate the relationships between asymmetric convection and vortex dynamics by using an intermediate complexity asymmetric framework, which lies between full-physics and balanced models.

Previous studies using unforced nondivergent barotropic models have provided a basic understanding that an annular ring of high PV is barotropically unstable. The mixing of PV leads to a stable configuration, resulting in weakened maximum winds. However, while unforced models can explain the axisymmetrization of asymmetries observed in nature, they are unable to explain why TCs can still undergo intensification in the presence of asymmetries. By necessity, intensification of the vortex requires some kind of forcing, which has often been through prescribed friction and diabatic heating in prior studies. Rozoff

et al. (2009) developed a forced barotropic model by parameterizing diabatic heating as a vorticity generation term in an annular region representative of an eyewall. They demonstrated that the barotropic instability and vorticity mixing act as a



temporary intensification brake, but continued forcing can lead to further intensification. Their results suggest that forced vorticity generation in the eyewall allows for the rebuilding of an annular vorticity ring and intensification.

Building upon the Rozoff et al. (2009) study, Hendricks et al. (2014) advanced the approach by using a forced shallow-water model that incorporates divergent effects and a parametrization of convective diabatic heating as a time-dependent annular mass sink with various widths. Their results demonstrate that a constant net heating prescribed in the region of high inertial stability inside the radius of maximum wind can lead to intensification. Hendricks et al. (2014) notes that "... when the heating is prescribed to be proportional to the relative vorticity, the maximum velocity increased and the minimum pressure decreased during barotropic instability...". Both Rozoff et al. (2009) and Hendricks et al. (2014) demonstrate that intensification can occur with sufficient forcing even in the presence of barotropic instability.

Schubert et al. (2016) presented another approach to parameterize diabatic heating in an axisymmetric shallow-water framework. Their approach involves parameterizing diabatic heating as a mass sink that is proportional to the fluid depth, resulting in a non-conservative contribution to the PV equation. They developed an analytic solution for TC intensification based on the wave-vortex approximation by Salmon (2014) and prescribed heating. In subsequent work, Schubert and Taft (2022) showed that vortex intensification in the shallow-water framework is proportional to the volume of mass removed from the vortex interior. In our current study, we employ a similar approach to parameterize diabatic heating as a mass sink. However, we introduce a key modification by allowing the forcing to be dynamically determined by the atmospheric flow. This modification enables us to incorporate the influence of the evolving atmospheric conditions into our analysis.

Organized moist deep convection originates from the TC boundary layer (TCBL) where the surface heat fluxes and radial inflow provide fuel for intensification and maintenance (Ooyama, 1969). Williams et al. (2013) used an axisymmetric slab TCBL model to show that nonlinear advection produces a shock-like structure with a sharp radial gradient of inflow just inside the radius of maximum wind (RMW), which leads to strong updrafts to initiate convection in TC. These results, along with Kepert and Wang (2001), Kepert (2017), and others have demonstrated that the location of strong updrafts is closely coupled to the boundary layer processes. A cloud resolving model simulation of Supertyphoon Haiyan (2013) conducted by Tsujino and Kuo (2020) shows that the coupling of a TCBL strong updraft and large vorticity could build a PV tower, which suggests that TCBL processes can play an important role in determining the location of heating, generating PV, and intensifying the TC.

To study how asymmetric boundary layer impacts the organization of convection and wind distribution of a TC, Shapiro (1983) is the first to use a depth-averaged, asymmetric slab boundary layer (SBL) model to investigate the role of the translation speed. The simplified model can qualitatively capture the features observed in nature; therefore Kuo et al. (2016) used a simple framework with a two-layer non-divergent barotropic model coupled with a slab boundary layer (SBL) to investigate why deep convection preferentially locates at the vertices of a polygonal eyewall. They used a time-varying pressure gradient forcing derived from the barotropic model to drive the winds in the SBL model. Their results show that pressure gradients associated with VRWs in the free atmosphere can organize the updrafts in the SBL, indicating that deep convection should be located close to the vertices, and rotate in concert with the VRWs. Although Kepert (2010) demonstrates that the SBL can produce overly strong inflow and too large a departure from the gradient balance, Kuo et al. (2016) findings shed new light on the dynamics of polygonal eyewalls we see in nature by reproducing quantitatively reasonable responses to the free atmospheric



forcing with the simplified boundary layer model. However, the two-layer model they employed still has some limitations in studying the effects of free atmospheric divergence and intensification. Their modeling setup allows one-way interaction only, such that the TCBL cannot influence the free atmosphere. Furthermore, the barotropic model is unforced and has no vorticity
source term, so the framework cannot be used to study TC intensification.

In this study, we develop a novel framework believed to capture the essential dynamics of polygonal eyewalls in an intensifying TC with a divergent shallow water model (SWM) on top of a SBL model to emulate a frictional boundary layer underneath a free atmospheric troposphere. We seek to investigate both how the deep convection is organized at the vertices and how the heating from the convection impacts TC intensification. The two-layer model maintains an approximate gradient wind balance
in the free atmospheric layer and parameterizes the diabatic heating produced by convection from the vertical motion out of the boundary layer. Our approach avoids prescribing the location and strength of the heating as has been done in previous studies. In our framework, the evolution of the free atmosphere and the heating from the TCBL are closely coupled. Section 2 describes the model and experiment design. Section 3 presents the results from the numerical experiments, and a discussion and summary is given in section 4.

## 2 Model and Experimental Design


A simple modeling framework of a shallow water model on top of an asymmetric slab boundary layer model is used in the study. The two-layer model is used in two different configurations denoted "one-way" and "two-way". The one-way model allows the pressure gradient force (PGF) from the shallow water layer to modulate the wind field in the slab boundary layer, but the boundary layer's flow cannot feed back to the upper layer. The two-way model allows for the two-way interaction, such
that the SBL can influence the free atmosphere through a parameterized mass sink. Details of the model design and the two configurations are shown below.

### 2.1 The one-way model

The SWM on a constant-$f$ plane solves the nonlinear shallow-watter equations of inviscid flow with no diffusion and no friction in a polar coordinate system:

$$\frac{\partial u}{\partial t} = -u\frac{\partial u}{\partial r} - v\frac{\partial v}{r\partial \lambda} + fv + \frac{v^2}{r} - \frac{\partial(gh)}{\partial r}, \tag{1}$$

$$\frac{\partial v}{\partial t} = -u\frac{\partial v}{\partial r} - v\frac{\partial v}{r\partial \lambda} - fu + \frac{uv}{r} - \frac{\partial(gh)}{r\partial \lambda}, \tag{2}$$

$$\frac{\partial h}{\partial t} + \frac{\partial(ruh)}{r\partial r} + \frac{\partial(vh)}{r\partial \lambda} = 0 \tag{3}$$

where $u$ is the radial velocity, $v$ is the tangential wind, $f$ is the constant Coriolis parameter set to 5.0 x $10^{-5}$ s $^{-1}$, and $h$ is the deviation of the fluid height from the mean depth $H$ in the SWM, which is set to a constant 2000.0 meters to represent a lower
tropospheric layer of the atmosphere. For the one-way model, the SBL cannot provide feedback to the SWM, ensuring the



material conservation of potential vorticity, denoted as $P = (f + \zeta)/h$. Here, relative vorticity $\zeta$ is defined as $\zeta = \partial(rv)/r\partial r - \partial u/r\partial\lambda$, and the material derivative is given by $D/Dt = \partial/\partial t + u(\partial/\partial r) + v(\partial/r\partial\lambda)$.

The governing equations for the symmetric dynamics in the boundary layer in polar coordinates, following Batchelor (1967); Kepert (2001); Shapiro (1983); Williams et al. (2013) and Williams Jr. (2015), are:

$$\frac{\partial u_b}{\partial t} = -u_b \frac{\partial u_b}{\partial r} - v_b \frac{\partial u_b}{r\partial\lambda} + w^- \left( \frac{u - u_b}{h_b} \right) + \left( f v_b + \frac{v_b^2}{r} \right) - g\frac{\partial h}{\partial r} - C_D U \frac{u_b}{h} + K \left( \nabla^2 u_b - \frac{u_b}{r^2} - \frac{2}{r^2}\frac{\partial v_b}{\partial\lambda} \right) \tag{4}$$

$$\frac{\partial v_b}{\partial t} = -u_b \frac{\partial v_b}{\partial r} - v_b \frac{\partial v_b}{r\partial\lambda} + w^- \left( \frac{v - v_b}{h_b} \right) - \left( f u_b + \frac{u_b v_b}{r} \right) + g\frac{\partial h}{r\partial\lambda} - C_D U \frac{v_b}{h} + K \left( \nabla^2 v_b - \frac{v_b}{r^2} + \frac{2}{r^2}\frac{\partial u}{\partial\lambda} \right) \tag{5}$$

$$w_b = -h \left( \frac{u_b}{r} + \frac{\partial u_b}{\partial r} + \frac{\partial v_b}{r\partial\lambda} \right), \tag{6}$$

$$\tag{7}$$

The radial wind is denoted by $u_b$ where the subscript $b$ indicates the boundary layer, $v_b$ is the tangential wind in the boundary layer, and $w_b$ is the diagnostic vertical wind at the top of the boundary layer. The 10-m wind speed used to calculate surface fluxes is

$$U = 0.78 \left( u_b^2 + v_b^2 \right)^{\frac{1}{2}} \tag{8}$$

and

$$w^- = \frac{1}{2} \left( |w_b| - w_b \right) \tag{9}$$

is the rectified Ekman suction, which results in a source term for the horizontal momentum in the boundary layer from downdrafts aloft. We opted to include both the Ekman suction term used in Williams et al. (2013); Williams Jr. (2015), as well as the full diffusion terms used in Kepert (2001) and Shapiro (1983) in this study, but sensitivity tests without these extra terms produced very similar results and are not shown here. A constant $C_D$ drag coefficient is set to $2.4 \times 10^{-3}$ following Bell et al. (2012a), the horizontal diffusion $K$ is set to $5000.0 \text{ m}^2 \text{ s}^{-1}$, and the height of the slab layer $h_b$ is set to a constant 1000.0 meters. Due to the presence of inertial oscillations in vertical motion that arise from the use of a slab boundary layer (Kepert, 2010), we used a stronger value of horizontal diffusion than previous studies rather than applying any special numerical filters. Smaller values of $K$ produced stable numerical integrations and qualitatively similar results in the axisymmetric and low wavenumber structure, but had apparent high wavenumber oscillations in the $w_b$. Increasing the $K$ value to $5000.0 \text{ m}^2 \text{ s}^{-1}$ largely removed the inertial oscillations and did not change the overall conclusions of the study.

In the one-way configuration, the two layers are coupled together via the SWM pressure gradient associated with gradients in the fluid depth, which remains close to gradient balance with the rotational wind in the shallow-water later in the absence of friction. The pressure gradient drives the SBL model and leads to sub- and supergradient winds, radial inflow, and vertical motion, due to the force imbalance from the surface drag and diffusion terms. The SBL model is therefore entirely driven by the free atmosphere dynamics from the pressure gradient terms and the nonlinear interactions of the boundary layer flow. The schematic of the one-way model configuration is shown in Fig. 1a.





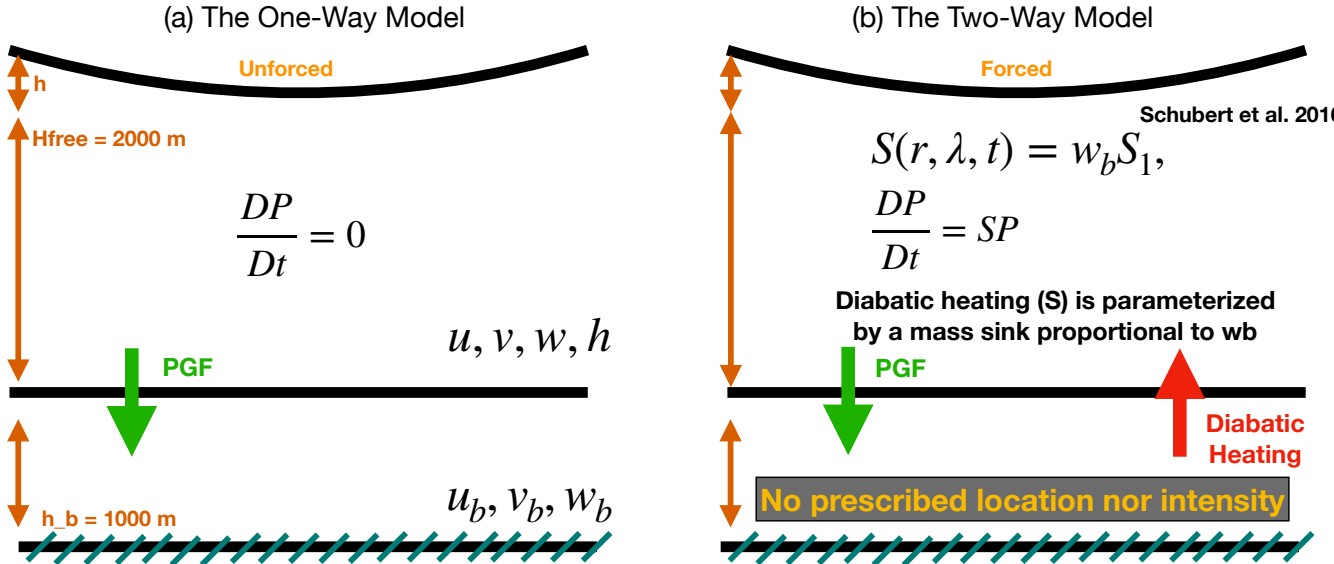

**Figure 1.** Schematics of the (a) One-way model and (b) Two-way model.

## 2.2 The two-way model

For the two-way model, we add a forcing term $S(r, \lambda, t)$ with units of s$^{-1}$ which simulates the effects of diabatic heating in a continuously stratified, compressible fluid. Unlike previous models that parameterized the effects of eyewall diabatic heating from a prescribed set of functions, here we allow the vertical motion to freely develop in response to the atmospheric and SBL

forcing.

We assume that the forcing is proportional to the vertical motion produced by convergence and divergence in the boundary layer.

$$S(r, \lambda, t) = w_b S_1 \tag{10}$$

where $S_1$ is a constant and set to $10^{-5}$ m$^{-1}$. A larger value of $S_1$ corresponds to more intense heating for a given $w$. Equation

3 now becomes

$$\frac{\partial h}{\partial t} + \frac{\partial (r u h)}{r \partial r} + \frac{\partial (v h)}{r \partial \lambda} = -hS \tag{11}$$

Because of the mass sink term $-hS$, the potential vorticity is no longer materially conserved, such that

$$\frac{DP}{Dt} = SP, \tag{12}$$

Therefore, $P$ in the free atmosphere can be generated by the updraft coming out of the boundary layer in a similar manner to

vertical gradients of heating in the Rossby-Ertel formulation in a stratified fluid. The relationship between the mass sink and





vertical gradient of heating can be understood through analogous forms of the mass continuity and potential density equations in the shallow water equations and isentropic coordinates (Schubert and Taft, 2022). In effect, the parameterization can be thought of as convective mass flux out of a lower tropospheric isentropic layer into the upper troposphere. We note that parameterizing heating as a mass sink in a single layer is a simplification that treats the upper tropospheric layer implicitly, and therefore

neglects entrainment and does not allow for any vertical structure in the heating profile. However, the simple parameterization acts in a similar way to deep convection in a TC as a PV source in the lower troposphere that can lead to intensity and structure change. The schematic of the two-way model configuration is shown in Fig. 1b. While mass is not exactly conserved in the shallow water layer in the two-way configuration, the fluid depth loss is many orders of magnitude smaller than the mean depth during the relevant time frame. Lack of strict mass conservation is not a problem for the length of time integration and the aims

of the study considered here, but the model is not expected to reach a steady state with this numerical approach.

### 2.3 Numerical model design

A new semi-spectral numerical weather prediction model was used in this study called the Scythe model (Bell and Cha, 2024). The model is formulated in polar coordinates using the spectral transform method (Orszag, 1970) to integrate the coupled shallow water and boundary layer equations forward in time. The spatial fields are discretized using finite elements in the

radial direction and Fourier series in the azimuthal direction on a reduced Gaussian grid. The finite elements are cubic b-splines (Ooyama, 2002) similar to the numerical representation in the SAMURAI variational analysis package (Bell et al., 2012b; Cha and Bell, 2023). The radial nodal spacing of the b-spline amplitude points is 3 km, with 3 Gaussian quadrature points in between each node that are spaced ∼1 km apart. One advantage of the finite elements over periodic basis functions is the ease of imposing boundary conditions. The boundary conditions are a Dirichlet zero wind condition at the origin and a

Neumann zero first derivative at the outer boundary for $u$ and $v$. A Neumann condition is set for $w$ and $h$ at the inner boundary, and no boundary condition is enforced for those variables at the outer boundary.

In the azimuthal direction, an evenly spaced grid is employed starting with 8 points around the circle at the radius closest to the origin. Four points are added at each successive radius, and are offset from the previous ring to form an approximate triangles in physical space. The grid arrangement is similar to that employed in the "octahedral" grid employed by the European

Center for Medium Range Weather Forecasting (ECMWF) global model where the spherical Earth is approximately projected onto an octahedron. In the current polar geometry, the grid is essentially a triangular tesselation with ∼1 km spacing between gridpoints throughout most of the domain, with slightly smaller spacing near the origin.

The time integration is similar to that described by Ooyama (2002). Here we use a third order Adams-Bashforth time integrator which provides better numerical stability and accuracy than the second order method (Durran and Blossey, 2012),

and the time tendencies are added in physical space rather than in spectral space due to the mix of b-spline and Fourier coefficients. At each time step, the spectral amplitudes of the Fourier modes and cubic b-spline elements are used to calculate the wind and fluid depth at each gridpoint in physical space in both layers. The spectral method allows for high accuracy in the calculation of the spatial derivatives in both the radial and azimuthal directions. Nonlinear advection terms and all other forcings are calculated in physical space and applied as time tendencies. The updated physical terms are then converted back



into spectral space with filtering and boundary conditions applied. The Fourier amplitudes are filtered according to the "cubic representation" such that the shortest wave on the grid is represented by 4 points. The b-spline amplitudes are filtered with the so-called "spline-cutoff" filter, which enforces a third derivative constraint on the spline function (Ooyama, 2002). The filtering and the spectral transform method eliminates spectral aliasing in the nonlinear terms.

The so-called "pole problem" associated with converging radials near the origin is handled through the combination of the
reduced Gaussian grid and the Fourier filtering. At the ring closest to the origin, only azimuthal wavenumber 1 is retained after Fourier filtering. Increasing wavenumbers are allowed at each ring as the number of points is increased radially. The model shows good numerical stability and accuracy in a wide variety of test cases and in the scientific results presented herein. The model code used in this study is open source and available for researchers interested in reproducing the results of this study.

## 2.4 Initial conditions

We use an initial vorticity profile that resembles an axisymmetric Rankine vortex and integrate the model for 3 hours to develop an approximately steady-state boundary layer flow. The initial profile is

$$V_T = V_{max} \left( \frac{R}{R_{max}} \right), \qquad\qquad R \leq R_{max}, \qquad\qquad (13)$$

$$V_T = V_{max} \left( \frac{R_{max}}{R} \right), \qquad\qquad R > R_{max} \qquad\qquad (14)$$

This idealized Rankine vortex possesses a circular rotation under the gradient wind balance with no transverse circulation.
$V_{max}$ is set to 50 m s$^{-1}$ and $R_{max}$ is set to 50 km, which are designed to be representative of a relatively intense. The mass field is initialized in gradient wind balance:

$$\frac{V^2}{r} + fV = g\frac{\partial h}{\partial r} \qquad\qquad (15)$$

We initialize this idealized profile for the two-layer model but allow one-way coupling only in the spin-up period. The pressure gradient force in the shallow-water model can therefore organize and develop the flow in the boundary layer, but
the flow in the boundary layer is not allowed to feed back into the shallow water during the spin-up phase. We then take the output at 3 hours, when the boundary layer flow is fully developed, as the initial conditions for subsequent experiments with asymmetries and two-way forcing. Figure 1 shows that a "shock" like structure of the tangential wind develops in the boundary layer, accompanied by an updraft up to 5 m s$^{-1}$ and radial inflow up to -24 m s$^{-1}$. Supergradient boundary layer winds are located inside the RMW of the flow aloft, and a sharp radial gradient of radial flow transitioning from inflow to
outflow around the $R_{max}$ suggests a convergence of radial momentum and an induced updraft that is consistent with the vertical velocity profile. The vorticity in the shallow water model (Fig. 1b) still closely resembles a Rankine profile and the wind field is in approximate gradient wind balance. In this simple initial condition, the mean vorticity is mainly 0.002 s$^{-1}$ inside the $R_{max}$ with small deviations due to the numerical integration. The mean vorticity radial gradient is near zero inside $R_{max}$ and exhibits a sharp negative radial vorticity gradient between 40 and 60 km, supporting the propagation of stable vortex



Rossby wave perturbations near the RMW. The flow does not have a vortex 'skirt', but rather a near zero radial gradient that damps outward radial propagation of VRWs. The fluid depth field shows the effective pressure gradient that drives the flow in the boundary layer.

**Figure 2.** The initial axisymmetric radial profile of the (a) wind flow in the SBL and tangential wind in the SWM, and the (b) vorticity, and shallow water depth deviation from the mean in the SWM.

After the initial spin-up of the model and SBL response, we then add a wavenumber 2 vortex Rossby wave (VRW) perturbation to the vortex following Lamb (1932) and Lee et al. (2006):



$$V_T = \frac{1}{2}\zeta R\left(\frac{\epsilon}{R_{max}}cos(2\lambda)\right), \qquad\qquad R \leq R_{eli}, \qquad (16)$$

$$V_T = -\frac{1}{2}\zeta\frac{R_{max}^2}{R}\left(\epsilon\frac{R_{max}}{R^2}cos(2\lambda)\right), \qquad\qquad R > R_{eli}, \qquad (17)$$

$$V_R = \frac{1}{2}\zeta R\left(\frac{\epsilon}{R_{max}}sin(2\lambda)\right), \qquad\qquad R \leq R_{max}, \qquad (18)$$

$$V_R = \frac{1}{2}\zeta\frac{R_{max}^2}{R}\left(\epsilon\frac{R_{max}}{R^2}sin(2\lambda)\right), \qquad\qquad R > R_{eli}, \qquad (19)$$

where $\epsilon$ is a amplitude factor set to 5 km, and $R_{eli} = R_{max} + \epsilon cos(2\lambda)$ is the elliptical boundary of maximum winds. This
perturbation superimposes a wavenumber two disturbance onto the Rankine vortex.

The magnitude of the tangential and radial wind of this wavenumber 2 VRW is the same but with a $\pi/4$ ($45^o$ degree)
difference across all radii. The wavenumber 2 disturbance manifests itself as two pairs of counter-rotating vortices. The total
circulation becomes an ellipse with the added wavenumber 2 disturbance. This profile with a balanced Rankine vortex and
added wavenumber 2 VRW is used as the initial condition for running subsequent one-way and two-way experiments. The
initial condition of a relatively intense TC and elliptical eyewall is designed to approximate the vortex structure observed during
the landfalls of Hurricane Michael (2018) (Cha et al., 2020) or Typhoon Herb (1996) (Kuo et al., 1999). After initialization,
the model is integrated forward for 24-hours to examine the vortex evolution.

## 3   Results

Time series of the one-way and two-way simulations of the maximum wavenumber 0 (axisymmetric) tangential velocity and
the radius of maximum wind in the SWM are shown in Figure 3. Since the SBL is not allowed to transport momentum upward
in the one-way model, and there are no other forcings, the maximum wavenumber 0 tangential wind in the SWM remains
mostly unchanged, with a magnitude oscillating between 43 and 47 m s$^{-1}$ due to the model's numerical integration and the
rotating elliptical asymmetry. The results suggest that the simulation reaches an approximate steady state in the axisymmetric
structure and intensity. The radius of maximum wavenumber 0 tangential wind mostly remains around 51 km, consistent with
our initial condition setup.

The two-way interaction simulation allows for the parameterized mass sink that is a proxy for diabatic heating and PV
generation to freely develop and influence the free atmospheric layer. The maximum wavenumber 0 tangential wind intensifies
throughout the simulation, but at a different intensification rate. The first time period bracketed by the black dashed lines has
an intensification rate of 0.9 m s$^{-1}$ hr$^{-1}$ after applying a 1-2-1 filter to reduce noise. The second time period bracketed by the
red dashed lines has a faster intensification rate of 1.5 m s$^{-1}$ hr$^{-1}$. Overall, the intensification from 50 to $\sim$80 m s$^{-1}$ is nearly
double the rapid intensification (RI) definition of the National Hurricane Center (NHC) of 30-knot (15.4 m s$^{-1}$) increase in a
tropical cyclone's maximum sustained winds within a 24-hour period (Kaplan and DeMaria, 2003). While we acknowledge the
physical discrepancy between the idealized tangential wind in the SWM compared to the 1-minute sustained wind used in the







**Figure 3.** Time series (in minutes) of (a) the maximum wavenumber 0 tangential velocity, and (b) the radius of maximum wind (RMW) for the shallow-water layer in the one-way and two-way simulations. The time periods of the different intensity rate stages are denoted by the dashed lines.



NHC definition, it is clear that the simplified vortex has rapidly intensified over the 24-hours of the simulation period. Higher
frequency oscillations in the wavenumber 0 tangential wind intensity can be seen in the time series between 400 and 800
minutes, suggesting the presence of asymmetries. After 800 minutes, there are less oscillations, and the wind speed intensifies
steadily. The purple dashed line denotes the time when the intensification rate slows down with a reappearance of an intensity
oscillation, suggesting that the structure becomes dynamical unstable again near the end of the simulation. Figure 3b shows
the evolution of the radius of maximum wind for the two simulations. The one-way simulation shows no change in the RMW,
consistent with the steady-state intensity. In the two-way simulation, the RMW contracts slowly with a value around 42 km.
When the intensification rate increases (period 2), the RMW contracts from 42 km to 30 km within 6.4 hr. The contraction rate
becomes slower after 1100 minutes and the RMW stays around 30 km while the tangential wind continues to intensify.

Figure 4 shows a time-radius diagram of the wavenumber 0 PV in the SWM and the wavenumber 0 vertical motion in the
SBL. The wavenumber 0 PV in the one-way model stays mostly unchanged with a monopole structure, and the wavenumber
0 updraft is located about 10 km inward of the RMW, consistent with Kepert (2017). The wavenumber 0 PV in the two-way
model starts from a monopole structure, and an enhanced PV ring appears between 25 and 35 km radius after 200 minutes.
The PV ring is collocated with the updraft in the SBL and continues to intensify and contract. The PV ring breaks down and
evolves into a monopole after 800 minutes, and an enhanced PV ring starts to reappear around 950 minutes and continues to
intensify. The transient slow-down of the intensification and subsequent reintensification are similar to Rozoff et al. (2009)'s
results.

Figure 5 illustrates a time-radius diagram of wavenumber 0 tangential and wavenumber 2 PV amplitude. In the one-way
experiment, the evolution of wavenumber 0 and wavenumber 2 amplitude are out of phase, with the asymmetry weakening
as the wavenumber 0 tangential wind intensifies. For example, around 800 minutes, when the tangential wind weakens, an
enhanced wavenumber 2 asymmetry appears. These reverse pulses of symmetric and asymmetric vortex amplification suggest
that the growth of the symmetric vortex may come at the expense of the decaying asymmetry and vice versa. However, the
growth and decay are a stable oscillation and neither amplitude departs far from the initial condition.

In contrast, during the two-way experiment the wavenumber 0 tangential wind begins intensifying at 200 minutes, whereas
in the one-way experiment, the vortex remains below 47.5 m s$^{-1}$. Despite both experiments being initialized with the same
wavenumber 2 VRW perturbation, an enhanced wavenumber 2 PV anomaly continues to grow and radially expand in the
two-way simulation until around 900 minutes before it begins to decay. During this time frame, the wavenumber 0 tangential
wind continues to intensify with some fluctuations in intensity, accompanied by the contraction of RMW. Unlike the one-way
simulation, the asymmetric and axisymmetric amplitudes increase together in-phase rather than out of phase.

After the denoted $7.5 \times 10^{-7}$ m$^{-1}$ s$^{-1}$ wavenumber 2 PV contour disappears, a second phase occurs where the mean tangen-
tial wind intensifies even more rapidly, reaching over 80 m s$^{-1}$ before the wavenumber two amplitude starts to increase again.
The results suggest a close coupling between the intensifying mean tangential wind and PV, indicating that the parameterized
PV forcing sustained by the boundary layer updraft can continuously generate PV in the free atmospheric layer over the entire
simulation period (Figs. 4 and 5).







**Figure 4.** Time radius diagram of the wavenumber 0 potential vorticity (shading) in the SWM and the wavenumber 0 vertical velocity (white contour with an amplitude of 2 m s$^{-1}$ in the SBL from the (a) One-way and (b) Two-way models. The yellow line denotes the RMW, and the dashed lines denote the time periods which are consistent with Fig. 3.



**Figure 5.** Similar to Fig. 4 but showing the wavenumber 0 tangential wind (shading) and $3.5 \times 10^{-7}$ and $7.5 \times 10^{-7}$ m$^{-1}$ s$^{-1}$ wavenumber-2 PV (black contour). Blue line denotes the RMW.





**Figure 6.** Similar to Fig. 4 but showing the wavenumber 0 radial vorticity gradient (shading).



The radial vorticity gradient from the one-way simulation (Fig. 6a) indicates that the vorticity exhibits a monopole-like pattern throughout the entire analysis period. In contrast, the two-way experiment reveals the emergence of a dipole pattern around 400 minutes, with positive radial gradient primarily located inside the 30 km radius and negative radial gradient occurring around 40 km. The change in sign of the radial vorticity gradient provides a necessary condition for barotropic instability. During period one between 400 and 800 minutes, there is a noticeable enhancement in the dipole radial vorticity gradient, which continues to around 900 minutes. The positive gradient becomes weaker at that time, and then the dipole gradient reintensifies around 1200 minutes, with the negative radial reaching an amplitude exceeding $4\times10^{-7}$ m$^{-1}$s$^{-1}$. The growth of the wavenumber two asymmetry closely follows the amplitude of the radial vorticity gradient (c.f. Figs. 5 and 6). The correspondence between the two is consistent with the idea that the asymmetry grows through barotropic instability associated with counter-propagating VRWs, and can also influence the axisymmetric vorticity gradient through vorticity mixing processes.

Figure 7 shows the time evolution of the asymmetric vertical motion and tangential wind in the SBL from 400 min to 592 min from the one-way model simulation. Similar to Kuo et al. (2016)'s finding, strong updrafts are found at the major axes of the ellipse, collocated with enhanced boundary layer supergradient tangential winds. By performing a spectral analysis similar to Cha et al. (2020)'s approach, the cyclonic rotation speed of the wavenumber 2 boundary layer updraft, tangential flow, and PV in the SWM are all 24.5 m s$^{-1}$, which takes 218 minutes for a complete rotation. The harmonic rotation between the boundary layer flow and the wavenumber 2 asymmetry in the SWM (not shown) suggests a close coupling between the location of the updrafts and the PV structure aloft.

Figure 8 shows the two-way model evolution during the first period, which is qualitatively similar to the one-way evolution. The upward motion in the boundary layer depicts an elliptical shape with maximum at the major axis, but the wavenumber two asymmetry is more pronounced and the eyewall is more compact compared to the one-way results. The boundary layer tangential wind collocates with the upward motion and rotate cyclonically together. The propagation speed of wavenumber 2 tangential and vertical motion in the boundary layer are both 30.72 m s$^{-1}$ and the mean radius of maximum wind is at $\approx 42.6$ km, which results in 145 minutes to complete a rotation. The rotation speed is faster than the one-way model results, potentially attributed to a storm of higher intensity and a smaller RMW.

To investigate the relationship between the reflectivity and updraft in the free atmosphere, Figure 9 shows the dynamic structure evolution, and the $h$ field refers to the pressure field. Firstly, the one-way model results show that an elliptical eyewall shape and is similar to the boundary layer upward motion pattern shown in Fig. 7 as the boundary layer flow is organized by the pressure gradient force. The 47 m s$^{-1}$ tangential wind is denoted by a blue contour, and has an opposite pattern compared to the boundary layer tangential wind. Kuo et al. (2016) also found a similar characteristic for the different tangential wind patterns between the boundary layer and the free atmosphere. They interpret the enhanced tangential wind at the minor axis of the ellipse through the tangential momentum equation neglecting the pressure gradient force term:

$$\frac{Dv}{Dt} = -\left(\frac{v}{r} + f\right)u \tag{20}$$

The radial inflow is largest at the major axis, which results in the strongest $Dv/Dt$ located at the major axis. Tangential advection downstream yields a maximum at the minor axis in the free atmosphere.







**Figure 7.** 192 minutes of evolution of the boundary layer updraft (shading) and 65 m s$^{-1}$ tangential wind (contour) from 400 to 592 min in the one-way simulation. The interval between the panels is 24 minutes.







**Figure 8.** Similar to Fig. 7 but for the two-way model.





**Figure 9.** Similar to Fig. 6 but showing the evolution with a 96-minute interval of (a) one-way model vertical motion (shading), 120 m SWM fluid deviation depth (black contour), and 47 m s$^{-1}$ tangential wind (blue contour); and (b) two-way model vertical motion (shading), 120 m SWM fluid deviation depth (black contour), and 54 and 57 m s$^{-1}$ tangential wind (blue contour).



The amplitude of diagnostic vertical velocity in the SWM is much less than $0.5 \text{ m s}^{-1}$, which is an order of magnitude weaker than the maximum boundary layer vertical motion. We note that the updraft from the SBL does not provide any momentum to the SWM in either the one-way or two-case configuration. The SBL updraft effect on the SWM in the two-way configuration

is solely through the mass sink term, and there is no upward interaction at all in the one-way case. The vertical velocity in the SWM has a dipole pattern but is not generally located at the edge of the elliptical vortex. The pattern is not consistent with the observed location of maximum reflectivity in elliptical eyewalls, but instead would be associated with a broad pattern of ascent and descent within the eye. Differences in the vertical velocity pattern between the SWM and SBL suggest that the boundary layer dynamics are primarily responsible for the enhanced reflectivity at the vertices, and not through the divergence

field above the boundary layer.

Figure 9b shows the upward motion, $h$, and 54 and $57 \text{ m s}^{-1}$ tangential wind speed contours in the SWM from the two-layer model results. The upward motion is not at the major axis, but instead, a $\pi/4$ phase behind the upward motion in the boundary layer at 400 min. In the SWM, the vertical motion is directly linked to the divergence field, and has about a $0.8 \text{ m s}^{-1}$ amplitude. While the pattern is more elliptical than the one-way case, the SWM vertical motion is still an order of magnitude

smaller than the motion in the SBL, similar to the one-way case.

Comparing the upward motion and the tangential wind pattern in the SWM (Fig. 9b) to the SBL (Fig. 8), the enhanced tangential signal does not collocate with the strongest upward motion. During the beginning of the period, the dipole secondary circulation is similar to the one-way results and is maximized at minor axis of the ellipse. The pattern becomes more complex as both the jet locations and secondary circulations rotate at different speeds, resulting in the tangential wind maxima near the

350 major axis at 592 min, which is different from the one-way model result.

The dynamical evolution in the boundary layer and the shallow water during the first period is accompanied by development of a PV ring, an increasing wavenumber 2 PV signal, and enhanced radial vorticity gradients. Figure 10 shows the evolution of the vertical velocity and tangential wind in the boundary layer during the second period. The vortex becomes more compact and the elliptical eyewall contracts to a smaller RMW compared to the first period. At 736 min, the maximum updraft and

355 tangential wind are mainly at the major axis. The elliptical shape evolves into more of a wavenumber 4 pattern at 784 min and becomes more circular at 1072 min with a ring of strong upward motion ring over $10 \text{ m s}^{-1}$.

Figure 11 shows the PV evolution in the SWM overlaid with the boundary layer updraft during the second period. At 736 min, the elliptical shape has four PV patches with a magnitude of $3.5 \times 10^{-6} \text{ m}^{-1}\text{s}^{-1}$ downstream of the major axis and hollow PV inside the eye. Some of the PV rotates around the vortex while some is mixed into the vortex center and outside the RMW.

At 1120 min, the nearly circular PV at the vortex center has a maximum over $3.6 \times 10^{-6} \text{m}^{-1}\text{s}^{-1}$. Because the boundary layer updraft continues to produce the mass sink and associated PV generation, the total PV after mixing is larger than the total PV before mixing (at 736 minutes, Eq. 12). Meanwhile, the mean vortex continues to intensify. The vorticity mixing process has been suggested to be a detrimental process in previous barotropic models, but with the aid of the PV forcing, the vortex was able to sustain itself and intensify after the mixing process, suggesting that the PV forcing outweighs the negative impacts of

mixing in this case.





**Figure 10.** Similar to Fig. 8 but for the period from 736 to 1120 minutes in the two-way model and 80 m s$^{-1}$ boundary layer tangential wind (black contour).







**Figure 11.** Similar to Fig. 10 but for the evolution of potential vorticity in the shallow water layer (shading) and 6 m s$^{-1}$ boundary layer upward motion (black contour) in the two-way model.



The evolution of growth and axisymmetrization of the asymmetries seen in Fig. 11 is consistent with the wavenumber 0 PV structure and the radial vorticity gradient during this period described earlier. After 1120 min, the wavenumber 0 tangential wind continues to steadily intensify with minimal asymmetries (Fig. 5). The wavenumber 2 PV signal reappears around 1250 min, and the intensification rate starts to slow down. Figure 12 shows two snapshots of the SWM PV and boundary layer updraft at 240 min (4 hours) and 1320 min (22 hours) respectively. The vortex has evolved to an elliptical shape again at 1320 min, but now the PV is stronger in the eye, and there are two PV maximum patches over $4 \times 10^{-6}\mathrm{m}^{-1}\mathrm{s}^{-1}$ are collocated with the strong updraft ring. Some evidence of high-wavenumber weak inertial oscillations in the $w$ field are apparent at this time due to the slab boundary layer equations. By comparing the two-way model PV fields at 240 minutes and 1320 minutes, it is evident that PV has undergone significant evoluti99on, with a substantial increase observed in both the eye and eyewall regions. On the other hand, the PV field in the one-way model stays almost the same for the entire evolution (Fig. 4). We reiterate that the only forcing in the SWM is through the mass sink term, which is driven by the boundary layer updraft. The location of the boundary layer updraft in turn is controlled primarily by the pressure gradient force aloft, which is associated with the structure of the PV field. The simulated evolution of the growth and decay of asymmetries during the intensification of the mean vortex and the growth of the PV ring suggests that the nonlinear, two-way interaction between the PV field aloft and the boundary layer is a critical aspect of TC intensification.

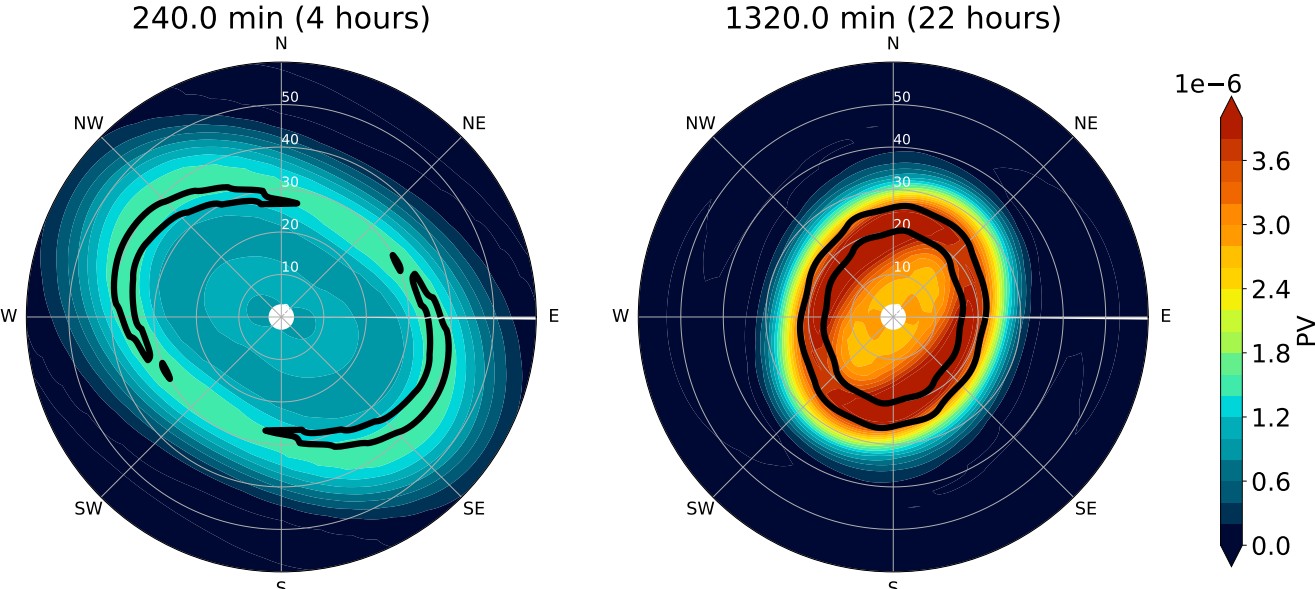

**Figure 12.** Similar to Fig. 11 but at 1320 minutes.





## 4 Conclusions

In this study, the relationships between vortex asymmetries, polygonal precipitation structure, and tropical cyclone (TC) intensification are explored through a simplified modeling framework. A two-layer model consisting of a shallow water model (SWM) representing the free atmosphere on top of a slab boundary layer (SBL) is used to understand the dynamical relationships. We conduct two experiments with a one-way and two-way configuration. These two experiments both use an approximate Rankine vortex with an added wavenumber 2 perturbation as the initial condition designed to mimic observed intense TCs with elliptical eyewalls. In the one-way configuration, the pressure gradient in the SWM layer drives the SBL, but no feedbacks are allowed to the free atmosphere. For the two-way interaction, we parameterize the effects of diabatic heating in the lower troposphere forced by boundary layer vertical motion as a mass sink, similar to the framework described in Schubert et al. (2016) and Schubert and Taft (2022). In the one-way configuration potential vorticity (PV) is materially conserved in the SWM, but in the two-way experiment the mass sink serves as a source of PV generation. Unlike previous studies with a prescribed heating profile (Rozoff et al., 2009; Hendricks et al., 2014; Schubert et al., 2016), the vertical motion in the boundary layer develops freely in these simulations, so the location and intensity of parameterized heating are determined by the flow field and changing at each time step, which is a more realistic yet still highly simplified heating distribution.

Results from the one-way experiment shows that the boundary layer updraft locates at the major axis of the ellipse, whereas the vertical velocity in the shallow water layer does not possess any elliptical shape. The vertical velocity in the boundary layer rotates cyclonically, and propagates in tandem with the PV signal in the free atmosphere. This result confirms Kuo et al. (2016)'s hypothesis and findings that the VRW in the free atmosphere can organize the updrafts in the SBL, such that the deep convection and enhanced radar reflectivity we see in nature is likely to be maximized at the polygonal vertices through this interaction.

Our one-way and two-way experiments further show that the vertical velocity derived from the shallow water layer is one order of magnitude smaller than the vertical velocity out of the boundary layer. The SWM has a maximum amplitude only up to 0.8 m s$^{-1}$ and does not have a consistent pattern with the observed maxima at the major axis, whereas the vertical motion in the SBL can be over 10 m s$^{-1}$ and is associated with the elliptical eyewall shape. A much weaker vertical motion in the SWM suggests that the convergence induced by the agradient flow aloft is weak compared to the convergence produced by frictional imbalance in the TCBL. The finding builds on Kuo et al. (2016)'s results which only had a barotropic layer aloft, and provides new evidence that the agradient forcing of convergence in the free atmosphere likely has little direct impact to produce observed polygonal precipitation structure. While the vertical motion in the SBL may be overestimated due to the use of a slab equation set instead of a height-resolved boundary layer, both the spatial pattern and the difference in magnitude from the SWM support the conclusion that the polygonal reflectivity pattern seen in radar observations is primarily due to the force imbalance between the pressure gradient force associated with vorticity asymmetries aloft and Coriolis, centrifugal, and frictional forces in the boundary layer. Our results also show that linear VRW theory is reasonable to describe the wavenumber 2 propagation in the one-way model simulation, similar to previous work. Similar results obtained from our one-way model and Kuo et al. (2016)'s barotropic free atmosphere layer suggest that a free atmospheric divergence is not required to capture



the essential dynamics of the observed coupling between vorticity and polygonal eyewall reflectivity (Kuo et al., 1999; Cha et al., 2020). However, further intensification of the mean vortex cannot occur in Kuo et al. (2016)'s experiment or the one-way experiments conducted here.

Results from the two-way experiment show that the parameterized PV generation driven by the boundary layer updraft can contribute to the growth of a PV ring and the mean vortex intensification. The PV generation term mimics the effects of
diabatic heating from deep convection through a mass sink in the shallow water layer. The results are consistent with Rozoff et al. (2009) and Hendricks et al. (2014)'s findings that parameterized diabatic heating can produce a strengthening PV ring, and that intensification can occur even in the presence of barotropic instability and eye-eyewall mixing. Hendricks et al. (2014)'s sensitivity experiment suggests the prescribed heating should be close to the center of the storm in order to intensify rapidly. Our model does not prescribe the heating location and intensity, but the free atmospheric flow organizes the heat source close
to the radius of maximum wind (RMW) through boundary layer processes. The location of the boundary layer updraft and associated mass sink is closely coupled with the contraction of the RMW and the building of the PV ring. The boundary layer updraft can develop freely in the model, but the flow itself determines the location to be near the PV maximum, so the PV grows nonlinearly and the vortex keeps contracting and intensifying. The cycling of the building of the PV ring and its breakdown are qualitatively similar to aircraft observations of intensifying storms (Kossin and Eastin, 2001). The evolution of the heating,
RMW, and PV ring are also broadly consistent with observational studies of Hurricane Patricia's (2016) rapid intensification (Martinez et al., 2019) and Hurricane Michael's (2018) rapid intensification (Cha et al., 2020).

The two-way experiment further shows that nonlinear interactions modulate the intensification rate, the growth and decay of asymmetries, and the growth and the breakdown of the PV ring throughout different stages of vortex evolution. Our result confirms Cha et al. (2020)'s finding that intensification of the mean vortex can occur even during barotropic instability, and
shows a similar evolution to Hurricane Michael's observations and analyses, including 1) growing low-wavenumber asymmetries accompanying by a reverse sign of the mean radial vorticity gradient, 2) axisymmetrization of asymmetries and a weakening mean radial vorticity gradient, and 3) a close coupling between the convection and polygonal vertices. The similarity between our model results and the observations suggests that the present modeling framework with two-way interactions captures many of the essential dynamics of TC rapid intensification and asymmetric structure. Given its simplicity, the frame-
work holds promise to provide further insight into key processes that control TC intensification with evolving asymmetries in the real atmosphere. We note that the current model does not have any latent heat fluxes or moisture at all which are known to be important in the real atmosphere, but the approximation of the dynamical effects of moist convection through PV generation is sufficient to intensify the vortex. Frictional imbalance in the boundary layer is essential to placing the PV generation at the right spot for this intensification to occur.

Future work should include sensitivity experiments to investigate a fuller exploration of the parameter space within this modeling framework, and how those choices either inhibit or facilitate intensification. Conducting sensitivity tests involving different initial conditions with varied radial wind profiles and perturbations to further explore the sensitivity to vortex structure is recommended. Additional experiments initializing the PV ring with varying initial conditions to assess the impact of different PV ring sizes on intensification and asymmetry evolution, and modifying factors such as the heating rate, Rossby deformation



length, boundary layer depth, and free atmosphere depth are expected to lead to further insight into TC intensification. The addition of a third layer could enable simulation of additional phenomena, such as entrainment and vertical wind shear. A three-layer asymmetric model would harken back to seminal work by Ooyama (1969) using an axisymmetric model, but with the added simulation of asymmetric structure and barotropic and baroclinic instabilities. Despite the simplicity of such models, they continue to yield new insights into the complexities of the real atmosphere.

*Code and data availability.*  The Scythe model code used in this study is written in the Julia programming language and is available at https://doi.org/10.5281/zenodo.10668054 and through M. Bell's GitHub page (mmbell).

*Author contributions.*  The study was designed by both authors. MMB built the model with contributions from TC. TC carried out the analysis and made the figures under the supervision of MMB. TC wrote the paper with contributions from MMB.

*Competing interests.*  The authors declare that they have no conflict of interest.

*Acknowledgements.*  This research was supported by the Office of Naval Research Tropical Cyclone Rapid Intensification Departmental Research Initiative and awards N00014-20-1-2069 and N00014-22-1-2737. This material is also based upon work supported by the NSF National Center for Atmospheric Research, which is a major facility sponsored by the U.S. National Science Foundation under Cooperative Agreement No. 1852977.



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
