# Peer review of "Tropical Cyclone Asymmetric Eyewall Evolution and Intensification in a Two-Layer Model"

_EGUsphere, 2024_

## Author Comment (AC1)

egusphere-2024-505

Reviewer #1:

The two-layer model (modeling boundary layer and free atmosphere), developed by the authors, can express the effect and feedback from the boundary layer to the free atmosphere. The effect and feedback cannot be expressed by the previous one-way configuration. The model has the potential to understand essential processes of the TC vortex development. The model is also suitable for the scaling law study. A scaling law is a power law that asserts a proportional relationship between relevant quantities. In general, the finding of the scaling law for a phenomenon in fluid systems can be largely helpful for understanding the phenomenon. It will be very fascinating to apply the two-way setting model to the scaling law studies in many different typhoon cases. Overall, the paper is well-written and can be published.

Thank you for reviewing the manuscript and providing constructive and helpful comments. We have made edits to the manuscript incorporated with your suggestions. Reviewers' comments are shown in black, our response to each comment is shown in blue, and changes to the manuscript are shown in red.

**Major**

My major concern with the paper is a potential gap between the authors' model and the real observations or full-physics model simulations. Specifically, the momentum transport in the two-layer model may need to be considered on some occasions. As the authors also pointed out, even in the two-way configuration, the updraft from the SBL does not provide momentum to the SWM (L332-340). That is, the vortex or tangential wind in the free atmosphere in the two-way configuration may be enhanced through gradient wind adjustment by the change in the pressure distribution of the free atmosphere associated with the updraft from the SBL. If so, there may be some differences in the intensifying process of the vortex between the authors' model and the full physics model. In the full physics model, the momentum transport associated with the updraft from the boundary layer to the free atmosphere greatly affects the development of the vortex in the free atmosphere (e.g., Fig. 6 of Wang et al. 2016). This suggests that the TC vortex can develop by a process that cannot be described by the authors' model. To what extent the evolution of the vortex represented by the authors' model is valid for the evolution of real TC vortices does not seem to be discussed in the current manuscript. If the authors can quantify or estimate the validity of their model, it may be useful to add a discussion in the text.

Reference: Wang, H., C. Wu, and Y. Wang, 2016: Secondary Eyewall Formation in an Idealized Tropical Cyclone Simulation: Balanced and Unbalanced Dynamics. JAS, 73, 3911-3930, https://doi.org/10.1175/JAS-D-15-0146.1

Thank you for this important and insightful point, and we agree with you that there are indeed caveats and limitations to this idealized model. We are fully aware of the importance of the vertical advection of momentum transport out of the boundary layer in the intensification process. Our intention with this simplified modeling framework is to distill the impacts on the vortex intensification process from diabatic heating, which is parameterized by a mass sink in this study. Our results do not preclude the possibility of other

mechanisms being important or even perhaps essential in the real atmosphere, and we have added new text to the manuscript to emphasize this important point. As we outlined in the introduction, while using a full-physics model simulation could aid in studying the kinematic and thermodynamic processes, the complexity involved makes it challenging to isolate and examine our proposed mechanism amidst the intricate interactions among various processes. We think that selective addition of various processes, such as vertical advection of momentum, could be very enlightening and a fruitful avenue for future work. In light of these considerations, we have incorporated an additional discussion in the manuscript to underscore the limitations of our model at L399:

We reiterate that the only forcing in the SWM is through the mass sink term, which is driven by the boundary layer updraft. The location of the boundary layer updraft in turn is controlled primarily by the pressure gradient force aloft, which is associated with the structure of the PV field. Our simplified modeling framework is specifically designed to distill the impacts on the vortex intensification process from diabatic heating forced by boundary layer convergence. However, such a simplification neglects other processes such as vertical advection of momentum out of the boundary layer, entrainment, vortex alignment and vertical growth, and other factors which impact the intensification process in the real atmosphere. We have identified one pathway to rapid intensification of a vortex in a highly simplified framework, but our results do not preclude the possibility of other mechanisms being important or even essential in the real atmosphere. For example, the advection of the boundary layer momentum and the adjustment of the supergradient winds in the boundary layer to gradient flow aloft is essential to prevent an eventual shock and frontal collapse in the shallow water layer. There is not a sufficiently strong supergradient flow to prevent the inward contraction of the RMW in the absence of that forcing, such that in the current framework, we must simply end the simulation before such effects become too strong to ignore. The slab boundary layer is in itself a simplification and likely overestimates the strength of the updraft in some areas, so additional momentum effects would need to be considered and modeled carefully. The current modeling framework provides an avenue for such additions of selective processes to design experiments with other intermediate levels of complexity. Such additions and experiments would enable a better assessment of which factors are indeed critical for intensification in the real atmosphere, and are recommended for future work.

**Minor**

There are some typographical errors in the model equations. Please correct them. For example, it may be necessary to revise the sign of the pressure gradient force in Eq. (5), u ($u_b$ correctly) in the last term of the right side of Eq. (5), and the specific expression of the Laplacian in Eqs. (4) - (5).

Thank you for catching the errors. We have corrected the terms and provided the expression of the Laplacian.

$$\nabla^2 = \partial^2/\partial r^2 + (1/r)(\partial/\partial r) + (1/r^2)(\partial^2/\partial \lambda^2)$$

---

## Author Comment (AC2)

Reviewer #2:

Referee: Rupert Klein
General comments:
The authors investigate the effect of asymmetric perturbations on the intensification of tropical storms based on a reduced two-layer model. The model consists of a slab boundary layer (SBL) and a superimposed layer with shallow water-type (SWM) dynamics. The SBL model is to represent the near-surface layer of the atmosphere influenced strongly by vertical turbulent transport, while the SWM represents about the lower 2-3km of the troposphere. The rest of the troposhere up to the tropopause is modelled implicitly by assuming that it has no dominant effect on vortex intensification and that it takes up and redistributes any vertical mass fluxes that may emerge out of the shallow water layer due to convection. These model components, including parameterizations of unresolved scale processes are adopted from the established literature, where they have already been used and argued for in similar contexts. In this sense, I consider the ingredients of this two-layer model to have stood the test of time as qualitative representations of some important aspects of large-scale atmospheric vortices. I do have some questions regarding the layer coupling, which I will post below.

The numerical scheme implemented to solve the model equations judiciously borrows from spectral and finite difference discretizations and is solidly state of the art.
The paper provides a detailed numerical study that juxtaposes model results with one- and two-way coupling of the two layers. In the one-way version, the SWM influences the SBL but not vice versa. The study clearly reveals that two-way coupling is crucial for reproducing key observed features of accelerating storms, such as a shrinking of the radius of maximum wind RMW during the intensification period, and – more importantly for the present paper – the interplay of Fourier mode one and two asymmetries during the process. Plausible physical interpretations of the processes observed in a series of model runs are provided, yielding an interesting set of hypotheses regarding the mechanisms behind what is called "rapid intensification" of tropical storms.
The paper is very well written, with a concise and clear literature review, well-structured technical descriptions of both the mathematical model used and of its numerical discretization, and with clear discussions of the simulation results.

Thank you for reviewing the manuscript and providing constructive comments. We have made edits to the manuscript incorporated with your suggestions. Reviewers' comments are shown in black, our response to each comment is shown in blue, and changes to the manuscript are shown in red.

**Specific Comments**

1. I do have one concern regarding the structure of the two-layer model. In lines 174, 175, the authors state that "Lack of strict mass conservation is not a problem for the length of time integration and the aims of the study considered here, but the model is not expected to reach a steady state with this numerical approach." I urge the authors to provide an extended argument leading to this conclusion for the following reason: The spin-up of a vortex is largely driven by the conservation of angular momentum and the fact that in the boundary layer mass is moving inwards, thereby inducing acceleration of the primary circulation. The inward-moving mass must, for conservation reasons, go somewhere. If I understand it

correctly, it is assumed here that the mass more or less slips through the SW layer and then disappears in the implicitly modelled bulk of the troposphere. What justifies assuming that the SW layer does not pick up at least part of that mass - an effect that would counteract that of the assumed "mass sink" attributed to convection and entrainment? And why would the implicitly modelled upper part of the atmosphere, which does absorb the upward mass flux and should, therefore, reveal a slow-down, not influence the shallow water layer at all?

Thank you for the insightful comment. You are correct that that the convective mass flux is continuously removed from the assumed mass sink during the numerical integration and that it implicitly must be deposited into the layer above. We have assessed the total amount of mass removed from the simulation in Fig. R1 to address the reviewer's concern.

The fluid depth is nearly constant for the one-way experiment as expected, with only small fluctuations due to the open boundary condition on $h$ and some very small dissipation from the model numerics. As the reviewer notes, the mass in the boundary layer can be considered to be constantly replaced through the radial inflow associated with the inward movement of the angular momentum surfaces. Mass continuity is prescribed exactly due to the diagnostic $w$ and the fixed height of the slab boundary layer. Any air that leaves the boundary layer is implicitly replaced by exactly the correct amount via radial inflow.

On the other hand, the mass from the shallow water layer in the two-way experiment was reduced around 27 %, which could be considered a substantial reduction in 24-hours. To address the reviewer's concern, we have incorporated an additional discussion in the manuscript at L376:

[Figure]

Figure R1: Evolution of the mass fraction removed from the shallow water layer over time.

A calculation of the fluid depth removed from the shallow water layer indicates that the amount is very small at any single time step or gridpoint (generally less than 0.1 mm), but that the cumulative effect is substantial and the total mass is reduced by $\sim 27\%$ of its original value. Our results are consistent with the conclusion of Schubert et al. (2016) that vortex intensification in the shallow-water framework is proportional to the volume of mass removed from the vortex interior. We have effectively assumed that the updraft erupting from the boundary layer is accelerating through the shallow water layer, such that the vertical motion at the top of the layer is larger than at the bottom and mass is removed. This assumption is consistent with a bottom-heavy convective mass flux profile in the lower troposphere. If the top of the layer is considered to be an isentrope, this the mass removal is

equivalent to a positive vertical gradient of diabatic heating across the layer. Implicitly then, the mass removed from the layer is being deposited into an unresolved upper-tropospheric layer where it must either accumulate or be removed through radial outflow.

In the simulated boundary layer the mass can be considered to be constantly replaced through the radial inflow associated with the inward movement of the angular momentum surfaces. Mass continuity is prescribed exactly due to the diagnostic $w$ and the fixed height of the slab boundary layer. Any air that leaves the boundary layer is implicitly replaced by exactly the correct amount via radial inflow. We can make such an implicit assumption in the unresolved upper-tropospheric layer as well, such that the mass deposited from the lower-layer is exactly ventilated by the radial outflow. Under that assumption, the upper-layer is entirely passive but maintains mass continuity in the full the atmosphere. Interestingly, it does not appear to be essential to explicitly simulate this layer to achieve rapid intensification, but addition of those effects could result in a change of the intensification rate. An additional third layer could allow for the development of an upper-level anti-cyclone and permit the inclusion of vertical wind shear and top-heavy stratiform mass flux profiles. A third layer would also allow for the effects of entrainment by considering the fraction of mass passed through the lower layer as in the axisymmetric simulations by Ooyama (1969). We leave such additions to the modeling framework to future work, but acknowledge that the specific intensification rate simulated here depends on the assumptions made about the mass flux profile and the passive role of the implicit upper-layer.

2. The authors report to impose homogeneous Neumann inner boundary conditions for vertical velocity, w, and boundary layer height, h. While I can see how that can be justified for the height from radial momentum balance, I don't see (i) why this condition should hold for w and (ii) why there should be a boundary condition for w in the first place. According to (6), the vertical velocity is the product of the boundary layer height and the horizontal divergence. Even if the height satisfies a homogeneous Neumann condition, I don't think the horizontal divergence would do so. If I am right, the radial gradient of w is height times the radial gradient of the divergence. Moreover, nowhere in the governing equations does the radial derivative of w occur, so why should a boundary condition for w be needed at all. What am I missing?

Thank you for carefully reviewing the manuscript and bringing this to our attention. Since the boundary layer updraft is diagnostically calculated, prescribing a boundary condition for $w$ is unnecessary. We have removed the statement and added an additional sentence to clarify this point.

$w$ is diagnostically calculated throughout the model integration over time.

**Minor comments:**

l. 113: shallow watter $->$ shallow water
l. 146: later $->$ layer
l. 374: evoluti99on $->$ evolution

We have corrected the typos as suggested. Thank you!